# Whole Exome-Wide Association Identifies Rare Variants in *GALNT9* Associated with Middle Eastern Papillary Thyroid Carcinoma Risk

**DOI:** 10.3390/cancers15174235

**Published:** 2023-08-24

**Authors:** Rong Bu, Abdul K. Siraj, Saud Azam, Kaleem Iqbal, Zeeshan Qadri, Maha Al-Rasheed, Saif S. Al-Sobhi, Fouad Al-Dayel, Khawla S. Al-Kuraya

**Affiliations:** 1Human Cancer Genomic Research, King Faisal Specialist Hospital and Research Center, P.O. Box 3354, Riyadh 11211, Saudi Arabia; rbu@kfshrc.edu.sa (R.B.); asiraj@kfshrc.edu.sa (A.K.S.); saazam@kfshrc.edu.sa (S.A.); miqbal@kfshrc.edu.sa (K.I.); sqadri96@kfshrc.edu.sa (Z.Q.); mrasheed@kfshrc.edu.sa (M.A.-R.); 2Department of Surgery, King Faisal Specialist Hospital and Research Center, P.O. Box 3354, Riyadh 11211, Saudi Arabia; sobhi@kfshrc.edu.sa; 3Department of Pathology, King Faisal Specialist Hospital and Research Center, P.O. Box 3354, Riyadh 11211, Saudi Arabia; dayelf@kfshrc.edu.sa

**Keywords:** papillary thyroid cancer, exome-wide association study, *GALNT9*, rare variants, sequence kernal association test

## Abstract

**Simple Summary:**

This study targeted the identification of rare variants in Middle Eastern papillary thyroid carcinoma (PTC) through an exome-wide association study. It was found that the *GALNT9* gene was strongly associated with rare inactivating variants. Three genes (*TRIM40*, *ARHGAP23*, and *SOX4*) were associated with rare damaging variants (RDVs). Furthermore, seven genes (*VARS1*, *ZBED9*, *PRRC2A*, *VWA7*, *TRIM31*, *TRIM40*, and *COL8A2*) were associated with PTC risk. The study unveils some new genes to be potential candidates for PTC predisposition.

**Abstract:**

Papillary thyroid carcinoma (PTC) is the commonest thyroid cancer. The majority of inherited causes of PTC remain elusive. However, understanding the genetic underpinnings and origins remains a challenging endeavor. An exome-wide association study was performed to identify rare germline variants in coding regions associated with PTC risk in the Middle Eastern population. By analyzing exome-sequencing data from 249 PTC patients (cases) and 1395 individuals without any known cancer (controls), *GALNT9* emerged as being strongly associated with rare inactivating variants (RIVs) (4/249 cases vs. 1/1395 controls, OR = 22.75, *p* = 5.09 × 10^−5^). Furthermore, three genes, *TRIM40*, *ARHGAP23*, and *SOX4*, were enriched for rare damaging variants (RDVs) at the exome-wide threshold (*p* < 2.5 × 10^−6^). An additional seven genes (*VARS1*, *ZBED9*, *PRRC2A*, *VWA7*, *TRIM31*, *TRIM40*, and *COL8A2)* were associated with a Middle Eastern PTC risk based on the sequence kernel association test (SKAT). This study underscores the potential of *GALNT9* and other implicated genes in PTC predisposition, illuminating the need for large collaborations and innovative approaches to understand the genetic heterogeneity of PTC predisposition.

## 1. Introduction

Thyroid cancer is the most common endocrine malignancy [1,2]. Papillary thyroid carcinoma (PTC) is a particular subtype that is responsible for a significant majority of all thyroid cancers, accounting for approximately 85% of patients [3,4]. The incidence of PTC has increased significantly in recent years [5,6]. In Saudi Arabia, PTC is of high prevalence and it is the second most common cancer affecting females [7]. Although PTC has favorable outcomes, 3–10% of patients demonstrated recurrent disease within the first decade after treatment [8,9].

Heritable genetic factors contribute to the susceptibility of PTC in around 10% of patients who have a family history of the disease in first or second-degree relatives and, in many instances, genetic effect has been shown to extend even beyond the nuclear family [10,11,12,13,14].

Despite the suggestive evidence that PTC is relatively highly heritable, understanding the genetic underpinnings and origins remains a challenging endeavor across diverse ethnic populations. Genetic studies, such as genome-wide association studies (GWAS), have offered some valuable insights, identifying several common genetic variants associated with PTC [15,16,17,18,19]. However, these identified variants account for only a modest portion of the total heritability of the disease, leaving a considerable proportion of the genetic architecture of PTC yet to be deciphered.

The advent of next-generation sequencing technologies such as whole-exome sequencing (WES) has been a powerful tool for uncovering the full spectrum of genetic variations that contribute to disease susceptibility. In contrast to GWAS, which primarily focus on common genetic variants, WES has the potential to uncover rare genetic variants that might play a significant role in various cancer susceptibilities [20,21,22]. These rare variants, which potentially are more deleterious than their common counterparts, might have been overlooked in previous studies due to their low frequencies. Therefore, WES has the potential to provide a more comprehensive picture of the genetic contributors to a complex disease like PTC, especially in a unique ethnicity where the prevalence of PTC is higher.

In this study, we harnessed the power of WES to identify rare inactivating variants (RIVs) and rare damaging variants (RDVs) associated with PTC risk within the Middle Eastern population. We hope our findings can shed light on the potential roles of several genes including *GALNT9* in conferring risk for PTC in this population, thereby contributing to a better understanding of this prevalent cancer.

## 2. Materials and Methods

### 2.1. Patient Selection

Two hundred and forty-nine PTC patients diagnosed between 2006 and 2018 at King Faisal Specialist Hospital and Research Center (Riyadh, Saudi Arabia) and Prince Sultan Military Medical City (Riyadh, Saudi Arabia), with fresh non-tumor tissue and peripheral blood available, were included in the study. Baseline clinico-pathological data were collected from case records and are summarized in Table 1. The histopathological subtypes of PTC were classified based on the fifth edition of the WHO classification for thyroid tumors [23]. The staging of PTC was performed using the eighth edition of the American Joint Committee on Cancer (AJCC) staging system [24]. Cases were identified based on clinical history followed by fine needle aspiration cytology for confirmation.

The Institutional Review Board and Research Advisory Council (RAC) of the King Faisal Specialist Hospital and Research Center approved this study under project RAC# 2211 168 and RAC# 2110 031. Patients provided written consent to be part of this study.

### 2.2. DNA Isolation

Genomic DNA was isolated either from fresh non-tumor tissues or peripheral blood utilizing a Gentra DNA isolation kit (Gentra, Minneapolis, MN, USA), following the manufacturer’s recommendations as described previously [25].

### 2.3. Exome Sequencing Analysis

Whole-exome sequencing for cases (*n* = 249) was carried out by utilizing the SureSelectXT Target Enrichment System (Agilent, Santa Clara, CA, USA) on the Illumina sequencing platform. The Burrows–Wheeler Aligner (BWA) v0.7.15 algorithm was used to align the sequencing reads to the human reference genome hg19, followed by the use of Picard tools (v1.119, http://broadinstitute.github.io/picard/, accessed on 2 March 2023) for local realignment and PCR duplication marking. The Genome Analysis Toolkit (GATK) v4.0.12.0 was utilized to perform base-quality recalibration. FastQC (http://www.bioinformatics.babraham.ac.uk/projects/fastqc/, accessed on 2 March 2023) and the GATK were used to obtain all quality metrics [26].

The GATK’s Haplotype Caller was used to call germline variants which were subsequently annotated using Annotate Variation (ANNOVAR) software (https://annovar.openbioinformatics.org/en/latest/, accessed on 2 March 2023) [27]. The control population included in our cohort was our in-house data from the exome sequencing of 1395 normal samples sequenced at different points in time. To provide homogeneity, all variants (cases and controls) were annotated simultaneously with ANNOVAR. Only those variants were selected that were in the exonic or splicing region of the gene. Variants were excluded if they had a minor allele frequency of >0.01 in the dbSNP, the National Heart, Lung, and Blood Institute exome sequencing project, 1000 Genomes, the Exome Aggregation Consortium (ExAC), and the Genome Aggregation Database (gnomAD). A variant was considered a true positive if the variant allele frequency (VAF) was at least 20% with a minimum altered reads of 5, and a sequencing depth in the variant location region of ≥20 with a GATK quality score of ≥100. Filtered variants were checked using the Integrated Genomics Viewer to filter out the false positives and artifacts.

Rare inactivating variants (RIVs) are deleterious variants that include frameshift insertion, frameshift deletion, stop-gain, stop-loss, and splice-site variants with allele frequencies < 1% in our control cohort and the ExAC database. On the other hand, rare damaging variants (RDVs) are the types of variants which are either deleterious/inactivating or predicted to be pathogenic/deleterious with a score greater than 0.025, utilizing the M-CAP classifier [21,28]. Furthermore, nonsynonymous variants include the variants with allele frequencies less than 0.1 in the ExAC database and our control cohort, including deleterious variants, missense single nucleotide variants, or non-frameshift deletions and insertions [21].

A Kyoto Encyclopedia of Genes and Genomes (KEGG) pathway analysis was performed, which is available on the Database for Annotation, Visualization and Integrated Discovery (DAVID) platform [29]. The gene list was uploaded with the default parameters for analysis. The results were considered statistically significant if *p* < 0.05.

### 2.4. Statistical Analysis

A chi square test was used to assess the departure from the Hardy–Weinberg equilibrium using a *p*-value < 0.001 as the cutoff. Variants were removed if they failed the Hardy–Weinberg equilibrium.

A sequence kernel association test (SKAT), which was implemented in an R package, was utilized for the association analysis. Standard parameters were used to compute the association. The *p*-values of non-synonymous variants were estimated using efficient resampling methods incorporated in the ‘SKATBinary_Single’ function. The exome-wide significance level was set at *p* < 2.5 × 10^−6^, whereas associations with *p* < 0.001 were labeled as suggestive [21].

To determine the power, simulation methods were applied which are available in the SKAT package under the ‘power logistic’ function. We repeated the simulation 100 times at different frequencies of variants in cases and odds ratios. The simulations were repeated to determine the power to detect a true association at the exome-wide (*p* < 2.5 × 10^−6^) and suggestive thresholds (*p* < 0.001). A power analysis was also calculated through a power simulation in R with varying samples, effect sizes, and threshold levels of significance.

## 3. Results

The median age of the entire cohort of cases was 39.2 years (range: 10–83 years), with a male/female ratio of 1:3.6. The majority of the tumors were of the classical variant (50.2%; 125/249), whilst 47.8% (119/249) of tumors were multifocal. Extrathyroidal extension was noted in 41.8% (104/249) of cases. Regional LN metastasis was noted in 38.2% (95/249) of cases and distant metastasis was present at diagnosis in 6.4% (16/249) (Table 1).

We detected 270,582 variants in 18,006 genes, including 91,701 RDVs (23,553 in cases and 110,797 in controls) and 13,743 RIVs (3114 in cases and 17,103 in controls). The cases and controls carried a median of 12 (interquartile range: 10–15) and 11 (interquartile range: 9–14) RIVs, respectively (Wilcoxon rank sum test *p* = 0.310).

The power analysis in our study was conducted utilizing the simulation-based procedure based on the variant distribution observed in our cohort. We found that our study had 80% power to identify a gene significantly enriched for RIVs with an odds ratio of 10 and a carrier rate of 7.5% at the exome-wide statistical significance level (*p* < 2.5 × 10^−6^). At the suggestive level (*p* < 0.001), our study had 80% power to detect an odds ratio of 10 and a carrier rate of 5% (Figure 1).

In the primary analysis focused on RIVs, *GALNT9* had the strongest association (adjusted odds ratio OR = 22.75, *p* = 5.09 × 10^−5^), meeting suggestive significance (2.5 × 10^−6^ < *p* < 0.001). RIVs were carried in 4/249 (1.6%) cases compared to 1/1395 (0.1%) control (Table 2). This association was driven by three different variants including chr12: 132690466A>G, a splicing variant observed in two cases and one control (OR = 11.28, *p* = 0.013). The other two variants were observed in one case each and were absent in controls (OR = 16.84, *p* = 0.018). (Table 3). No other gene besides *GALNT9* had suggestive evidence (*p* < 0.001) of an association with RIVs.

Furthermore, we utilized the filter-based approach to inspect the top 10 genes selected by counting the number of RIVs in the cases, which ranged from five RIVs in *TEKT5* to eight in *DNAH14*. When comparing the cases with controls, eight genes had significant evidence of association (*p* < 0.05), including *CYP4X1* (*p* = 0.004), *RBM23* (*p* = 0.020), *CRYGN* (*p* = 0.007), *TTC23L* (*p* = 0.030), *CYP2R1* (*p* = 0.030), *AP3S1*, (*p* = 0.002), *TTC23* (*p* = 0.005), and *TEKT5* (*p* = 0.005) (Table 4).

In addition, a KEGG pathway analysis was performed for signaling pathway analysis. No pathways were identified to be significantly associated with an increased risk of developing PTC.

In our secondary analysis focused on RDVs in the cases and controls, three genes passed the exome-wide significance level of association with RDVs, including *TRIM40* (*p* = 6.31 × 10^−9^), *ARHGAP23* (*p* = 1.05 × 10^−8^), and *SOX4* (*p* = 1.80 × 10^−7^) (Table 2). In these three genes, the associations were caused by multiple RDVs, each present in only one or three individuals. The exception was Chr6:30114877C>T in *TRIM40*, which was present in 5/249 cases and absent in the controls (OR = 62.78, *p* = 1.15 × 10^−7^) (Table 3).

As part of the above exploratory analyses, SKAT analysis was utilized to analyze the cumulative effect of all nonsynonymous variants with an allele frequency of less than 1% in our cohort. Seven genes, valyl-tRNA synthetase 1 (*VARS1*), Zinc finger BED-type containing 9 (*ZBED9*), Proline Rich Coiled-Coil 2A (*PRRC2A*), Von Willebrand Factor A Domain Containing 7 (*VWA7*), Tripartite Motif Containing 31 (*TRIM31*), Tripartite Motif Containing 40 (*TRIM40*), and Collagen Type VIII Alpha 2 Chain (*COL8A2*), had exome-wide evidence of association (Table 5). All seven genes harbored at least one individual variant with *p* < 0.001. Five individual variants had an exome-wide significant association after Bonferroni Correction (Appendix A).

## 4. Discussion

Most genetic susceptibility association studies on PTC to date have focused mainly on common variants, and the majority have also focused on the European population rather than other Asian populations, especially Middle Eastern. However, the proportion of the missing heritability in PTC could be explained by low-frequency rare variants.

Our study underscores the valuable role of whole-exome sequencing in uncovering the genetic underpinning of PTC risk. We shed light on the presence of rare inactivating variants (RIVs), primarily in the *GALNT9* gene, which according to our analysis had the strongest association with PTC in our research population.

The *GALNT9* gene, as a part of the N-Acetylgalactosaminyl transferase (*GALNT*) gene family, is involved in the initial stages of O-linked protein glycosylation in the golgi apparatus [30,31]. Thus, inactivating variants in the *GALNT9* gene, as detected in our study, could disrupt normal glycosylation processes and potentially contribute to oncogenic pathways. However, the precise role of *GALNT9* in cancer biology, particularly in PTC, remains under-explored and needs to be the focal point of future research to elucidate the potential role of *GALNT9* in cancer susceptibility.

Previous evidence indicates that mutations in *GALNT12*, another member of the N-Acetylgalactosaminyl transferase gene family, might explain some of the familial CRC cases of unknown etiology, and it showed that deleterious variants in *GALNT12* are associated with CRC susceptibility [32,33]. In addition, several prior studies have also demonstrated the implications of other GALNT family members in different cancer types [34,35,36,37,38,39,40]. This evidence lends credence to our findings and strengthens the hypothesis that *GALNT9* may have a similar role in PTC.

The enrichment of rare damaging variants (RDVs) in other genes such as *TRIM40*, *ARHGAP23*, and *SOX4* further complements our understanding of the heritable genetic landscape of PTC. Each of these genes have previously been implicated in various cancers, including gastrointestinal cancers, soft tissue sarcomas, osteosarcoma and prostate cancer [41,42,43,44,45,46,47,48,49]. However, there are a lack of reports on the roles of the *TRIM40*, *ARHGAP23* and *SOX4* genes in the development of PTC.

Interestingly, in our attempt to collapse the rare variants into pathways across the genome and look for their associations with risk of PTC, we failed to identify any pathway being significantly associated with increased risk of PTC, using KEGG pathway analysis.

While our study provides an essential step toward comprehending the genetic basis of PTC, it is vital to acknowledge the challenges posed by the genetic heterogeneity of this disease. The rarity of identified variants such as the *GALNT9* gene in this unique ethnic population demands larger collaborative efforts to substantiate their roles in PTC risk. The development and application of advanced statistical and bioinformatics methodologies could facilitate more robust genetic associations containing larger study cohorts, which could eventually help in the discovery of more risk-associated variants.

## 5. Conclusions

In conclusion, our study found novel genes and rare variants in *GALNT9*, *TRIM40*, *ARHGAP23*, and *SOX4* to be associated with PTC in the Middle Eastern population, suggesting potential biomarkers for PTC that are unique to this ethnicity. However, larger-scale studies are warranted to validate these findings due to the relatively small number of cases tested.

## Figures and Tables

**Figure 1 cancers-15-04235-f001:**
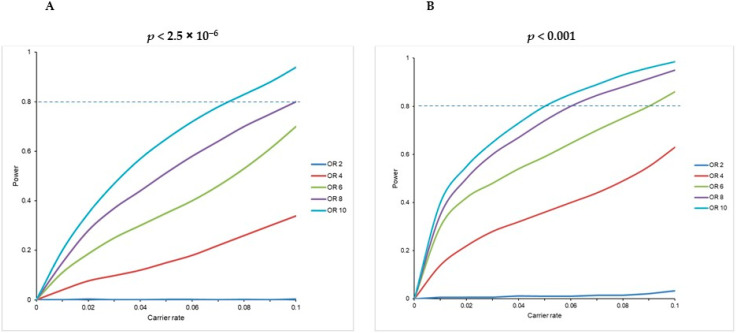
Power analysis for a gene to be enriched in rare inactivating variants based on an empirical procedure. (**A**) Power at the exome-wide significance level: *p* < 2.5 × 10^−6^. (**B**) Power at the suggestive significance level: *p* < 0.001.

**Table 1 cancers-15-04235-t001:** Clinico-pathological characteristics of papillary thyroid carcinoma cases.

	Total	
	No.	%
**Total**	249	
**Age at surgery (years)**		
Median (range)	39.2 (10.0–83.0)	
<55	210	84.3
≥55	39	15.7
**Gender**		
Male	54	21.7
Female	195	78.3
**Histologic subtype**		
Classical variant	125	50.2
Follicular variant	53	21.3
Tall cell variant	36	14.5
Other variants	35	14.0
**Tumor laterality**		
Unilateral	180	72.3
Bilateral	69	27.7
**Tumor focality**		
Unifocal	130	52.2
Multifocal	119	47.8
**Extrathyroidal extension**		
Absent	145	58.2
Present	104	41.8
**Lymphovascular invasion**		
Present	95	38.2
Absent	154	61.8
**Tumor size**		
≤1 cm	23	9.2
1.1–2 cm	45	18.1
2.1–4 cm	85	34.1
>4 cm	96	38.6
**Regional LN metastasis**		
N0	118	47.3
N1	95	38.2
Nx	36	14.5
**Distant metastasis**		
Present	16	6.4
Absent	233	93.6
**TNM Stage**		
I	217	87.5
II	20	8.1
III	6	2.4
IV	5	2.0

**Table 2 cancers-15-04235-t002:** List of genes significantly associated with PTC risk based on rare variant analyses.

S. No.	Gene	No. of Cases	%	No. of Controls	%	Odds Ratio	*p*-Value	Significance Level	Type of Variant Analysis
1	*GALNT9*	4	1.6	1	0.1	22.75	5.09 × 10^−5^	suggestive	RIV
2	*TRIM40*	6	2.4	0	0.0	74.50	6.31 × 10^−9^	exome-wide	RDV
3	*ARHGAP23*	7	2.8	1	0.1	40.32	1.05 × 10^−8^	exome-wide	RDV
4	*SOX4*	6	2.4	1	0.1	34.40	1.80 × 10^−7^	exome-wide	RDV

**Table 3 cancers-15-04235-t003:** List of rare variants significantly associated with PTC risk.

Gene	Chr	Position	Reference	Alternate	Variant Type	No. of Cases	%	No. of Controls	*%*	*p*-Value	Odds Ratio	Type of Variant
*GALNT9*	12	132,839,102	G	-	Frameshift deletion	1	0.4	0	0.0	0.018	16.84	RIV
*GALNT9*	12	132,688,237	-	GCGGGGAGACGGC	Splicing	1	0.4	0	0.0	0.018	16.84	RIV
*GALNT9*	12	132,690,466	A	G	Splicing	2	0.8	1	0.1	0.013	11.28	RIV
*TRIM40*	6	30,114,877	C	T	missense	5	2.0	0	0.0	1.15 × 10^−7^	62.78	RDV
*ARHGAP23*	17	36,667,177	C	T	missense	3	1.2	0	0.0	4.07 × 10^−5^	39.63	RDV
*SOX4*	6	21,595,904	C	T	missense	3	1.2	0	0.0	4.07 × 10^−5^	39.63	RDV
*ARHGAP23*	17	36,666,646	C	T	missense	1	0.4	0	0.0	0.018	16.85	RDV
*ARHGAP23*	17	36,614,410	C	T	missense	1	0.4	0	0.0	0.018	16.85	RDV
*ARHGAP23*	17	36,619,098	C	T	missense	1	0.4	0	0.0	0.018	16.85	RDV
*ARHGAP23*	17	36,622,633	C	T	missense	1	0.4	0	0.0	0.018	16.85	RDV
*SOX4*	6	21,595,481	C	T	missense	1	0.4	0	0.0	0.018	16.85	RDV
*SOX4*	6	21,595,276	G	A	missense	1	0.4	0	0.0	0.018	16.85	RDV
*SOX4*	6	21,595,618	G	A	missense	1	0.4	0	0.0	0.018	16.85	RDV
*TRIM40*	6	30,113,787	G	A	missense	1	0.4	0	0.0	0.018	16.85	RDV
*ARHGAP23*	17	36,619,383	G	A	missense	0	0.0	1	0.1	0.673	1.86	RDV
*SOX4*	6	21,595,955	C	G	missense	0	0.0	1	0.1	0.673	1.86	RDV

**Table 4 cancers-15-04235-t004:** Top 10 genes by number of cases based on RIV association analysis.

S. No.	Gene	No. of Cases	%	No. of Controls	%	*p*-Value	Odds Ratio
1	*DNAH14*	8	3.2%	45	3.2%	0.991	1.00
2	*WDR67*	7	2.8%	18	1.3%	0.071	2.21
3	*CYP4X1*	6	2.4%	8	0.6%	0.004	4.28
4	*CRYGN*	6	2.4%	9	0.6%	0.007	3.80
5	*RBM23*	6	2.4%	11	0.8%	0.020	3.11
6	*TTC23L*	6	2.4%	12	0.9%	0.030	2.85
7	*CYP2R1*	6	2.4%	12	0.9%	0.030	2.85
8	*AP3S1*	5	2.0%	5	0.4%	0.002	5.51
9	*TTC23*	5	2.0%	6	0.4%	0.005	4.59
10	*TEKT5*	5	2.0%	6	0.4%	0.005	4.59

**Table 5 cancers-15-04235-t005:** Significant genes based on non-synonymous variant associations analyzed using SKAT.

S. No.	Gene	No. of Cases	%	No. of Controls	%	*p*-Value (SKAT)	Odds Ratio
1	*VARS1*	11	4.4	0	0.0	3.98 × 10^−10^	134.58
2	*ZBED9*	8	3.2	0	0.0	1.26 × 10^−7^	98.23
3	*PRRC2A*	7	2.8	0	0.0	8.50 × 10^−7^	86.32
4	*VWA7*	7	2.8	0	0.0	8.50 × 10^−7^	86.32
5	*TRIM31*	7	2.8	0	0.0	8.50 × 10^−7^	86.32
6	*TRIM40*	7	2.8	0	0.0	8.50 × 10^−7^	86.32
7	*COL8A2*	7	2.8	0	0.0	8.50 × 10^−7^	86.32

## Data Availability

The data presented in this study are available in tables and figures of this manuscript.

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
