# Peer review of "Whole Exome-Wide Association Identifies Rare Variants in GALNT9 Associated with Middle Eastern Papillary Thyroid Carcinoma Risk"

_cancers, 2023, doi:10.3390/cancers15174235_

Round 1
Reviewer 1 Report
Well done article based on exome sequencing data, identifies potential high-risk SNPs for future validation studies. Limitations include the relatively small number of cancer cases tested and a lack of validation data to imply the significance of the finding.
Author Response
We thank the reviewer for their valuable time and encouraging comments. We appreciate the reviewer’s concern and acknowledge the limitations of our study. A validation study is indeed warranted to further elucidate the importance of these genes in PTC.

Reviewer 2 Report
The authors employed an exome-wide association study to uncover rare genetic variants that influence the development of inherited PTC. Although the methodology seems fine, the Introduction and Discussion section should provide more information and a broader understanding of the problem in question.
1. The heritability of PTC should be stated in the Introduction section. Is the heritability different in Middle East countries compared to global population? The authors state that heritable genetic factors contribute to the susceptibility of PTC in around 10%. However in the next sentence, they also state that PTC is highly heritable. 10% is not high heritability and suggest that other factors (environmental or other) contribute to the development of PTC rather than the genetic background.
2. The authors should explain the meaning of rare inactivating variants and rare damaging variants. I belive most readers are not familiar with these terms.
3. The abbreviations SNNOVAR and GATK should be explained
4. The legend of Table 1 is missing
5. Although it is clear that the authors investigated rare gene variants, the number of cases in Table 2 is very small. Can conclusions realy be drawn from these data?
6. Table 5 is unclear. What do the percentages mean?
7. The possible role of TRIM40, ARHGAP23 and SOX4 in PTC development should be commented
Minor English editing is needed
Author Response
Comments and Suggestions for Authors
The authors employed an exome-wide association study to uncover rare genetic variants that influence the development of inherited PTC. Although the methodology seems fine, the Introduction and Discussion section should provide more information and a broader understanding of the problem in question.
We appreciate the reviewer for their constructive comments. Keeping in view the reviewer’s recommendations, the manuscript has now been revised accordingly.
- The heritability of PTC should be stated in the Introduction section. Is the heritability different in Middle East countries compared to global population? The authors state that heritable genetic factors contribute to the susceptibility of PTC in around 10%. However in the next sentence, they also state that PTC is highly heritable. 10% is not high heritability and suggest that other factors (environmental or other) contribute to the development of PTC rather than the genetic background.
We thank the reviewer for their suggestion to further highlight the heritability in Middle Eastern countries compared to global population. Although there is no data available describing the incidence of hereditary PTC in the Middle Eastern population, it is expected to be higher than the global population owing to the high rate of consanguinity in this population. [1]
We agree with the reviewer, that 10% incidence of heritability is not very high and that other factors contribute to the development of PTC rather than the genetic background. Hence, we have modified the sentence as follows:
“Despite, the suggestive evidence that PTC is relatively highly heritable” (Page 2)
- The authors should explain the meaning of rare inactivating variants and rare damaging variants. I belive most readers are not familiar with these terms.
We appreciate the reviewer for their concern. Respecting reviewer’s suggestion, we have revised this part of material and methods as follows:
“Rare inactivating variants (RIVs) are deleterious variants including frameshift insertions, frameshift deletions, stopgain, stoploss, and splice-site variants with allele frequencies < 1% in our control cohort and the ExAC database. Whereas rare damaging variants (RDVs) are the type of variants which are either deleterious/inactivating or predicted to be pathogenic/deleterious with score greater than 0.025 utilizing the M-CAP classifier.” (Page 3)
- The abbreviations ANNOVAR and GATK should be explained:
We thank the reviewer for pointing this out. Full forms of these terminologies have now been added to the main text and the manuscript has been revised accordingly.
Page 3: “Genome Analysis Toolkit (GATK) v4.0.12.0 was utilized to perform base-quality recalibration”.
Page 3: “The GATK’s Haplotype Caller was used to call germline variants and subsequently annotated by Annotate Variation (ANNOVAR) software”.
- The legend of Table 1 is missing
The legend (title) of Table 1 was presented at page 3 as below:
“Clinico-pathological characteristics of papillary thyroid carcinoma cases”
- Although it is clear that the authors investigated rare gene variants, the number of cases in Table 2 is very small. Can conclusions realy be drawn from these data?
We appreciate the reviewer’s concern and we agree that the number of cases is small in our study. We have revised the conclusion in manuscript as below”
“In conclusion, our study found novel genes and rare variants, in GALNT9, TRIM40, ARHGAP23 and SOX4 to be associated with PTC in Middle Eastern population suggesting potential biomarkers for PTC that are unique to this ethnicity. However, larger scale studies are warranted to validate these findings due to the relatively small number of cases tested.” (Page 8)
- Table 5 is unclear. What do the percentages mean?
We thank the reviewer for their comment. The percentage in the tables indicates the frequency of cases or controls harboring variants in indicated genes in our study.
- The possible role of TRIM40, ARHGAP23 and SOX4 in PTC development should be commented
We thank the reviewer for their suggestion regarding the possible role of TRIM40, ARHGAP23 and SOX4 genes.in PTC development. The association of TRIM40, ARHGAP23 and SOX4 genes with development of PTC are still undetermined although abovementioned genes were reported to be involved in carcinogenesis of many cancers including gastrointestinal cancers, soft tissue sarcomas, osteosarcoma and prostate cancer [2-4] Therefore the manuscript has been revised as below:
“Each of these genes has previously been implicated in various cancers, including gastro-intestinal cancers, soft tissue sarcomas, osteosarcoma and prostate cancer. However, there is a lack of report on the role of TRIM40, ARHGAP23 and SOX4 genes in development of PTC.” (Page 7)
Minor English editing is needed
Author reply: The manuscript has been revised accordingly.
References
- AlHarthi, F.S.; Qari, A.; Edress, A.; Abedalthagafi, M. Familial/inherited cancer syndrome: a focus on the highly consanguineous Arab population. NPJ genomic medicine 2020, 5, 3.
- Noguchi, K.; Okumura, F.; Takahashi, N.; Kataoka, A.; Kamiyama, T.; Todo, S.; Hatakeyama, S. TRIM40 promotes neddylation of IKKgamma and is downregulated in gastrointestinal cancers. Carcinogenesis 2011, 32, 995-1004, doi:10.1093/carcin/bgr068.
- Sadaf, A.; Szabo, S.; Ferguson, K.; Sorger, J.I.; Sumegi, J.; Bridge, J.A.; Pressey, J.G. Novel ARHGAP23‐FER fusion in a metastatic spindle cell–predominant neoplasm with a myofibroblastic phenotype and a sustained metabolic response to lorlatinib. Cancer 2021, 127, 4124-4130.
- Sun, Z.; Zhang, T.; Chen, B. Long non-coding RNA metastasis-associated lung adenocarcinoma transcript 1 (MALAT1) promotes proliferation and metastasis of osteosarcoma cells by targeting c-Met and SOX4 via miR-34a/c-5p and miR-449a/b. Medical science monitor: international medical journal of experimental and clinical research 2019, 25, 1410.

Round 2
Reviewer 2 Report
The authors have improved the manuscript sufficiently for publication.